# Molecular Characterization of a B Cell Adaptor for Phosphoinositide 3-Kinase Homolog in Lamprey (*Lampetra japonica*) and Its Function in the Immune Response

**DOI:** 10.3390/ijms232214449

**Published:** 2022-11-21

**Authors:** Mengqi Chai, Xiujia Liu, Lin Wei, Jun Li, Meng Gou, Ting Zhu, Yinglun Han, Xin Liu

**Affiliations:** 1Lamprey Research Center, College of Life Science, Liaoning Normal University, Dalian 116081, China; 2Collaborative Innovation Center for Key Technologies of Marine Food Deep Processing, Dalian Polytechnic University, Dalian 116034, China

**Keywords:** *Lampetra japonica*, BCAP, Lipopolysaccharide, lymphocyte, immune response

## Abstract

Human B cell adaptor for phosphoinositide 3-kinase (BCAP) is identified as an adaptor protein expressed in B cells and plays a critical immunomodulatory role in B cell receptor signaling and humoral immune response. In the current study, a homolog of BCAP (Lja-BCAP) was identified in *Lampetra japonica*. The open reading frame of *Lja-BCAP* contains 2181bp nucleotides and encodes a protein of 726 amino acids. After being stimulated by mixed bacteria, the mRNA and protein expression levels of Lja-BCAP and the activation levels of tyrosine kinases increased significantly in peripheral blood lymphocytes, gills and supraneural myeloid bodies, respectively. However, after the knockdown of *Lja-BCAP* by RNAi in vivo, the activation of tyrosine kinases was inhibited in the above tissues, which indicated that Lja-BCAP participated in the anti-bacterial immune response of lampreys. After lipopolysaccharide (LPS) stimulation, the expression of Lja-BCAP in peripheral blood lymphocytes, gills and supraneural myeloid bodies were significantly up-regulated 2.5, 2.2, and 11.1 times (*p* < 0.05) compared to the control group, respectively; while after phytohemagglutinin (PHA) stimulation, the up-regulation of Lja-BCAP was only detected in peripheral blood lymphocytes. The above results show that Lja-BCAP mainly participates in the LPS-mediated immune response of lampreys.

## 1. Introduction

Adaptor protein is a type of protein that is very important in cell signal transduction. It can specifically interact with other signal proteins and activate downstream signal pathways, thereby activating transcription factors in the nucleus to regulate the transcription and expression of functional genes [1]. Human B cell adaptor for phosphoinositide 3-kinase (PI3K) (BCAP) was first characterized as an adaptor protein in B cells. Studies have found that in mice with BCAP deletion mutations, the amount of mature B cells is reduced, B1 cells are defective, the concentrations of serum immunoglobulin (Ig) M and IgG3 become low and the immune response to lipopolysaccharide (LPS) is weakened [2]. After the B cell receptor (BCR) binds to the ligand, BCAP-deficient B cells in mice showed reduced Ca^2+^ mobilization and a weakened proliferation response. These findings indicate that BCAP is a critical immunomodulatory molecule in B cell development and humoral immune response [3]. The BCAP protein sequence has five conserved domains (a Dof/BCAP/BANK (DBB) domain, an ankyrin repeat-like domain, a coiled-coil domain and two proline-rich domains) and an immunoreceptor tyrosine activation motif (ITAM) between the ankyrin repeat-like and the proline-rich domains [4]. The phosphorylation of tyrosine residues on ITAM of BCAP is mediated by Bruton’s tyrosine kinase (Btk) and spleen tyrosine kinase (Syk), thereby providing a binding site for the p85 subunit of PI3K [5]. When the BCR signaling is triggered, the ITAMs of Igα (CD79a) and Igβ (CD79b) of the BCR complex were phosphorylated by Lck/Yes-related novel protein tyrosine kinase (Lyn). The phosphorylated Igα and Igβ recruit and activate Syk, which further activates the B cell linker (Blnk) and Btk to initiate cascade reactions downstream of nuclear factor kappa B (NF-κB) and mitogen-activated protein kinase (MAPK) signaling pathways, respectively [6]. In addition, Castello et al. found that a non-ITAM tyrosine residue (Tyr204) of Igα was also phosphorylated by Syk, which induces the binding of Igα C-terminal with the Src homology (SH) 2 domain of an SH2-domain containing noncatalytic region of tyrosine kinase (Nck) [7]. The SH3 domains of activated Nck could interact with the proline-rich domains of BCAP. Then, the tyrosine phosphorylation on YxxM of the ITAM of BCAP can recruit the p85 subunit of PI3K, by activating its downstream protein kinase B (Akt), which participates in regulating B cell development, proliferation and differentiation [8,9].

In addition to being proven to play an important role in the BCR signaling pathway, recent studies have also revealed that BCAP seems to be involved in T cell receptor (TCR) signaling through the recruitment and activation of PI3K. Deason et al. found that BCAP was a key factor in the IL-1R-activated T cells by activation of PI3K/Akt signaling, which plays a key regulatory role in the differentiation of inflammatory Th17 cells [10]. Another study found that, after T cell activation BCAP was up-regulated with the initiation of PI3K signaling pathway, which continued to enhance PI3K signaling and promote the continuous transformation of cluster of differentiation 8 (CD8) positive T cells into memory and effector T cells. Whereas, the knockout of BCAP in CD8^+^ T cells will weaken the proliferation and development of effector and memory T cells after infection with *Listeria monocytogenes* [11]. BCAP is expressed after CD4^+^ T cell activation and is rapidly phosphorylated and interacts with the p85 subunit of PI3K during early TCR signaling, indicating that BCAP is involved in the recruitment and activation of PI3K in CD4^+^ T cell primordial cells, similar to its role in BCR signal pathway [12]. There is evidence that BCAP has also shown divergent activities in the maturation or transition of several innate immune cell lines such as macrophages and dendritic cells, etc. [13,14,15].

Based upon BCR, TCR, and the major histocompatibility complex, lampreys (jawless vertebrates) do not have an adaptive immune response. However, lampreys have been found to form an alternative adaptive immune system in another way. Pancer et al. found a new class of variable lymphocyte receptors (VLR) in the sea lamprey (*Petromyzon marinus*) [16]. The amino acid sequence of each type of VLR molecule is highly conserved at their N-terminal and C-terminal, but there are highly variable regions in its middle part, which are composed of leucine-rich repeat (LRR) modules [17]. Due to the rich diversity of LRR module sequences, after permutation and combination, a huge diversity of antigen recognition receptor molecules required by the adaptive immune system can be generated, thereby forming an alternative type of adaptive immune system [18]. Thus, for different types of antigens, each lymphocyte can produce only one kind of VLR molecule with specific structure [19,20,21].

There are three kinds of VLRs in lamprey; they are VLRA, VLRB and VLRC, which specifically express on three lymphocyte subsets, respectively [16,22]. VLRA^+^ and VLRC^+^ lymphocyte subsets develop and mature in the thymus-like tissue at the tip of the gills, and have immune responses to T cell mitogen stimulation [23]; the VLRB^+^ lymphocyte subsets (accounting for 90% of the total number of peripheral blood lymphocytes) develop in the supraneural myeloid bodies on the upper part of the spine, and have an immune response to B cell mitogen stimulation, and differentiate into plasmacytoid large lymphocytes, which can produce VLRB multimeric antibodies [24]. According to their immune response characteristics, VLRA^+^, VLRB^+^ and VLRC^+^ lymphocyte subsets are considered to be similar to the αβ T cell, B cell and γδ T cell subsets in higher vertebrates [23,24].

In recent years, although the VLRB antibody structure [25] and the assembly mechanism [26] of lamprey and hagfish have been deeply understood, the molecular mechanism of the adaptive immune response mediated by lymphocytes against pathogen infection in the lamprey immune system, and the molecular mechanism of VLRs’ signal transduction has not been clarified. In this study, we described the cloning, recombinant expression, antibody preparation and functional analysis of a BCAP homolog in *Lampetra japonica* (Lja-BCAP). This project is the preliminary analysis of the functional characterization of the role that Lja-BCAP plays in the immune response.

## 2. Results

### 2.1. Identification of Lja-BCAP and Sequence Similarity Analysis

The cDNA sequence of the Lja-BCAP was successfully cloned by the Nested PCR method using the primers listed in Appendix A. It contains an open reading frame (ORF) of 2181 bp, encoding a protein of 726 amino acids (aa) with a molecular weight of about 98 kDa. The multiple sequence alignment of Lja-BCAP and BCAPs of human, chicken, alligator, chimaera fish and sea lamprey were analyzed by using BioEdit7.0 software (Figure 1).

The results showed that the amino acid sequence of Lja-BCAP shares as high as a 90.6% identity with that of the sea lamprey, but two lamprey homologs are about 35.2–36.8% identical to BCAP sequences of those higher vertebrates. Though the overall sequence identities among them are not high, lamprey BCAP homologs also possess all five conserved functional domains and an ITAM motif which are the typical characteristics of BCAP family, which indicated that BCAP homologs exist in lampreys.

### 2.2. Phylogenetic and Conserved Motif Analyses of BCAP and BANK1 Families

In order to further understand the phylogenetic relationship of BCAP molecules, the phylogenetic tree was reconstructed by Neighbor-Joining (NJ) method with 45 BCAP and B cell scaffold protein with ankyrin repeats 1 (BANK1) sequences from 25 species which are representatives of six vertebrate classes and three invertebrate classes (Figure 2A, Appendix A). The results showed that the vertebrates’ BCAPs and BANK1s were grouped into two distinct large clusters; Lja-BCAP and BCAP of sea lamprey were clustered into the same cluster with those BCAP molecules of jawed vertebrates. An uncharacterized protein LOC116950877 from the sea lamprey which is about 17% identical to sea lamprey BCAP but possesses a DBB motif was clustered between BCAP and BANK1 groups. The genetic distance between protein LOC116950877 and BANK1s’ ancestor was closer than that between protein LOC116950877 and BCAPs’ ancestor. The BCAP-like from the echinoderms (*Acanthaster planci*) and the mollusks (*Crassostrea gigas*) and the BANK1-like of the coelenterates (*Hydra vulgaris*) are grouped as outgroups in a single clade, respectively (Figure 2).

The conserved motif analysis revealed that among the total predicted 20 conserved motifs (Appendix A), 10 motifs (1, 2, 3, 5, 6, 7, 8, 9, 13 and 17) are shared by BCAPs and BANK1s, two motifs (15 and 16) are unique to BANK1 and six motifs (4, 10, 11, 12, 14, 18) are unique to BCAP. Lamprey BCAPs share only one unique motif (15) with BANK1s, but share three unique motifs (4, 11 and 12) with BCAPs. In addition, Lamprey BCAPs share a motif combination (13-3-4-1-12) which is only conserved in the BCAP family, while protein LOC116950877 possesses another motif combination (13-3-1) that is only conserved in the BANK1 family. The results of phylogenetic and conserved motif analyses suggest that BCAP and BANK1 molecules have a common ancestral gene and diverged at least after the emergence of jawless vertebrates.

### 2.3. Induced Expression of Recombinant Protein and Preparation of Its Rabbit Polyclonal Antibody

The 200 aa at the C-terminal of Lja-BCAP was predicted to be an epitope rich fragment (Appendix A, Appendix A). The recombinant expression plasmid of truncated Lja-BCAP was constructed and transformed into *E. coli* Rosstta (DE3) competent cells, induced by Isopropyl β-D-thiogalactoside (IPTG), and then detected by sodium dodecylsulphate polyacrylamide gel electrophoresis (SDS-PAGE) (Appendix A). The results showed that the specific target band appeared at 40kD, which was consistent with the expected target protein size, and 90% of rLja-BCAP was expressed in the supernatant, indicating that the solubility of rLja-BCAP was high (Appendix A). The rLja-BCAP was purified by His-tag affinity chromatography to obtain a single-band pure recombinant protein (Appendix A).

The purified rLja-BCAP was used as the antigen, and the rabbits were immunized according to the immunization method described in Materials and Methods. After four booster immunizations, the rabbit anti-rLja-BCAP polyclonal antibody was successfully prepared. After the antibody was purified by affinity chromatography, the antibody titer was measured to reach 1:128,000 by ELISA detection (Appendix A). The specificity of the antibody was detected by WB analysis. The results showed that the rabbit anti-Lja-BCAP polyclonal antibody can specifically detect the rLja-BCAP induced by IPTG (Appendix A).

### 2.4. Functional Characterization of Lja-BCAP in Response to Mixed Bacteria Stimulation by RNAi Technique

To verify the function of *Lja-BCAP* in immune response, the RNA interference (RNAi) method was used to knockdown *Lja-BCAP* mRNA expression as described in Materials and Methods. The transcription and protein expression of *Lja-BCAP* were examined by quantitative real-time PCR (qPCR) and Western blotting methods, respectively.

According to the experimental results in Figure 3A,C, *Lja-BCAP* mRNA is constitutively expressed in immune related tissues, and the background expression of *Lja-BCAP* mRNA in peripheral blood lymphocytes and gills is higher than that in supraneural myeloid bodies. In the negative control group that was treated with scrambled Lja-BCAP small interfering RNA (siRNA), the transcription levels of *Lja-BCAP* increased significantly in the peripheral blood lymphocytes and gills but not significantly in the supraneural myeloid bodies after the mixed bacteria stimulation relative to the control group. In the RNAi group that was treated with Lja-BCAP siRNAs, the expression levels of *Lja-BCAP* mRNA were significantly depressed in peripheral blood lymphocytes and the supraneural myeloid bodies and slightly depressed in the gills relative to the negative control group after the mixed bacteria stimulation (Figure 3A).

The Western blotting (WB) results clearly showed that the Lja-BCAP expressions after mixed bacteria stimulation were all significantly knocked down in the immune-related tissues of the RNAi group relative to their corresponding control or negative control groups. However, in the negative control group, the Lja-BCAP expression levels after mixed bacteria stimulation were slightly up-regulated in the gills but significantly in the peripheral blood lymphocytes and the supraneural myeloid bodies relative to those of the control group (Figure 3B,C). Further, an anti-phosphotyrosine monoclonal antibody 4G10 was used to check the tyrosine phosphorylation levels in these immune-related tissues after mixed bacteria stimulation. It can be seen in Figure 3B that the tyrosine phosphorylation levels after mixed bacteria stimulation were up-regulated in all these immune-related tissues in the negative control group compared to the control group. However, after the knockdown of *Lja-BCAP*, the tyrosine phosphorylation levels were all blocked in these tissues after mixed bacteria stimulation compared to the control and negative control groups. The above results all prove that *Lja-BCAP* participates in the anti-bacterial immune response of lamprey.

### 2.5. Lja-BCAP Mainly Participates in the LPS-Mediated Immune Response of Lampreys

In order to further find out what kind of lymphocyte mitogen has an effect on Lja-BCAP expression, B cell mitogen LPS and T cell mitogen phytohemagglutinin (PHA) were used to stimulate the lamprey and the differential expressions of Lja-BCAP were examined in the immune-related tissues. The WB results showed that after 24 h of LPS stimulation, the expressions of Lja-BCAP in the peripheral blood lymphocytes, gills and supraneural myeloid bodies were 2.5, 2.2 and 11.1 times higher than that of the control group (*p* < 0.05), respectively. After PHA stimulation, the significant up-regulation of Lja-BCAP was only detected in peripheral blood lymphocytes by 3.5 times that of the control group, respectively. No differential expression was detected in the supraneural myeloid bodies, but the Lja-BCAP was significantly down-regulated in the gills (*p* < 0.05) (Figure 4A,B). The above results indicate that Lja-BCAP mainly participates in the LPS-mediated immune response of lampreys.

## 3. Discussion

Adaptive immunity is the specific response of lymphocytes to antigens under stimulation, which can produce immune memory effects and play a key role in completely eradicating pathogens and preventing re-infection [27]. Although jawless vertebrates do not have an adaptive immune system mediated by TCR, BCR and MHC such as in higher vertebrates [28], they have been found to use three types of VLR receptors to form another type of adaptive immune system. This has provided us with a key clue to reveal the origin and evolution of the vertebrate adaptive immune system [18]. In recent years, we have conducted in-depth explorations on the molecular mechanism of immune response in the immune system of *L. japonica*. We have discovered that some molecules related to the early events of the BCR signaling pathway in higher vertebrates are also involved in the immune response of lamprey. These molecules include Btk [29], Blnk [30], Syk [31], Vav guanine nucleotide exchange factor 3 (VAV3) [32] and Lyn [33]. These findings indicate that although the lamprey lymphocyte receptors and their possible coupling transmembrane molecules are different from higher vertebrate B cells, those molecules involved in intracellular signal transduction have a common genetic basis compared with higher vertebrate B cells.

In the present work, a homologous molecule of BCAP was also cloned in the lamprey. By searching through several genome databases, it was found that there are two BCAP homologous sequences in the sea lamprey (*P. marinus*). One (accession number: AY152674 [34]) is highly identical (>90%) to Lja-BCAP; though the identities among lamprey BCAPs and several typical BCAPs of high vertebrates are not high in their full sequence, their sequences are highly conserved in five domains that are typical characteristics of BCAP family. Another one named uncharacterized protein LOC116950877 (accession number: XM_032969002) is only 17% identical to sea lamprey BCAP (Appendix A). The sequence alignment result showed that protein LOC116950877 shares a little bit higher sequence similarity with BCAPs than with BANK1s of high vertebrates (Appendix A). However, the conserved motif analysis revealed that protein LOC116950877 shares a conserved motif combination (13-4-1) that only exists in the BANK1 family of jawed vertebrates (Figure 2). Given that no sequence could be found in the lamprey genome with high homology to vertebrates’ BANK1 in these genomic databases available, we deduced that protein LOC116950877 should be a BANK1 homolog in lamprey. Our results suggest that the divergence of BCAP and BANK1 molecules might have happened already in jawless vertebrates.

Because BCAP is an adaptor protein that can couple upstream BCR or TCR signaling to the downstream PI3K/AKT signaling in the immune reaction of higher vertebrates, our functional characterization of *Lja-BCAP* was first carried out to investigate what will happen after *Lja-BCAP* was knocked down by RNAi. After 48h *Lja-BCAP* siRNAs treatment and 24 h mixed bacteria stimulation, although there are no significant repressive effects on *Lja-BCAP* transcription in peripheral blood lymphocytes and gills in RNAi group relative to the control group (Figure 3A), the expression levels of Lja-BCAP were all down-regulated significantly (*p* < 0.01) in these three immune-related tissues of RNAi group compared with those of control group (Figure 3B,C). In addition, after 48 h of scrambled siRNA treatment and 24 h of mixed bacteria stimulation, the expression levels of *Lja-BCAP* mRNA and protein were nearly all up-regulated significantly in these immune-related tissues of the negative control group compared with those of the control group (Figure 3A–C). These results indicate that *Lja-BCAP* was successfully knocked down by the RNAi method.

The successful knockdown of Lja-BCAP expression provides the possibility for further exploring its function in the lamprey immune response. The significant up-regulation of *Lja-BCAP* protein was detected in the peripheral blood lymphocytes (*p* < 0.01), gills (*p* < 0.05) and supraneural myeloid bodies (*p* < 0.05) after mixed bacteria stimulation in the negative control group, and these up-regulations were blocked after *Lja-BCAP* knockdown (Figure 3B,C). Our findings at protein level were coincident with the results of Hirano et al. at the transcriptional level. They performed transcriptomics on VLRA^+^, VLRB^+^, TN and VLRC^+^ lymphocytes and found that BCAP is expressed in all of them, with the strongest expression found in VLRB^+^ lymphocytes [35]. In higher vertebrates, the early events after BCR and TCR activation are associated with tyrosine phosphorylation and dephosphorylation of Src-family protein tyrosine kinases [36]. The phosphorylation of two tyrosine residues on ITAM of BCAP is also critical for the activation of the PI3K p85 subunit [9]. Therefore, tyrosine phosphorylation is considered one of the key steps in BCR or TCR signal transduction in vertebrates [37]. Thus, a Mouse Monoclonal Antibody 4G10 was applied to check the tyrosine phosphorylation levels after the knockdown of Lja-BCAP [38]. In the negative control group, the tyrosine phosphorylation levels in peripheral blood lymphocytes, gills and supraneural myeloid bodies after mixed bacteria stimulation are obviously higher than those in the control group, and they were all attenuated after knockdown of *Lja-BCAP* (Figure 3B). Thus, it can be concluded that Lja-BCAP is functionally involved in the lamprey anti-bacterial immune response.

It was found that VLRA^+^ and VLRC^+^ lymphocytes develop in the thymus-like tissue distributed on the tip of lamprey gills and they can be activated by PHA [22]. LPS can stimulate the proliferation of lymphocytes in the peripheral blood and supraneural myeloid bodies [32]. Therefore, in order to find out what kind of mitogen Lja-BCAP is probably associated with, the expression profiles of Lja-BCAP were tested in these immune-related tissues under LPS and PHA stimulation. After LPS stimulation, the expression levels of Lja-BCAP are up-regulated extremely significantly (*p* < 0.01) in peripheral blood lymphocytes, gills and supraneural myeloid bodies (Figure 4A,B). Conversely, after PHA stimulation, the expression levels of Lja-BCAP were significantly up-regulated only in peripheral blood lymphocytes (*p* < 0.01), and did not change obviously in supraneural myeloid bodies but were down-regulated significantly in gills (*p* < 0.05), respectively. Yamazaki et al. found that mouse BCAP^−/−^ B cells lost 50% of the ability to respond to LPS stimulation compared to wild-type B cells [2]. From this point, it seems that Lja-BCAP is mainly associated with LPS-mediated immune reactions as in its vertebrate homologs.

Nevertheless, there are still some questions that need to be addressed in future studies. VLRB^+^ lymphocyte subsets (B-like cells) distribute in all of these immune-related tissues, and it is easy to connect Lja-BCAP with VLRB signaling. So, in order to obtain some hints between BCAP and VLRB signaling, VLRB positive or negative cells should be sorted out by high-affinity antibody for comparison. Similarly, the relationships of Lja-BCAP with VLRA or VLRC signaling should be verified as well.

## 4. Materials and Methods

### 4.1. Lamprey Immunization

Lamprey (*L. japonica*) were purchased from Tongjiang City, Heilongjiang Province, and then reared at the Lamprey Research Center of Liaoning Normal University. The healthy adults of lamprey were divided into a control group, mixed bacteria, LPS and PHA (Sigma, St. Louis, MO, USA) stimulation groups. Two aquatic bacteria *Streptococcus agalactiae* and *Vibrio parahemolyticus* were chosen as the representatives of Gram-positive (G^+^) and Gram-negative (G^−^) bacteria for preparing inactivated antigens, respectively. They were inactivated by acetaldehyde first and then mixed together in normal saline at a concentration of 1 × 10^7^ CFU/mL for each strain. LPS or PHA powder were dissolved in phosphate-buffered saline (PBS) buffer (0.01 M, pH 7.4), respectively, at a concentration of 1 mg/mL. After 1 week of acclimatization in a constant temperature water tank, lampreys in each group were immunized with a corresponding antigen solution (100 μL for each) through intraperitoneal injection according to the previous description [30]. The peripheral blood lymphocytes were separated from the blood according to Han et al. [30] by Mouse Lymphocyte Separation Medium (Haoyang, Tianjin, China), and other immune-related tissues such as gills, supraneural myeloid bodies were dissected and placed in RNA protective agent (Solarbio, Beijing, China) or tissue cryopreservation solution [30], and stored in ultra-low temperature refrigerator at −80 °C.

### 4.2. PCR Amplification of the Target Gene

A human BCAP protein sequence was used as query to search aligned sequences through BLASTP program against a lamprey database of our laboratory [39], a full-length cDNA sequence of *Lja-BCAP* was obtained, and primers were designed using Primer 5.0 software (Appendix A) for the next step of amplifying the ORF of Lja-BCAP. The supraneural myeloid bodies of *L. japonica* were isolated and total RNA was extracted with RNAiso Plus (Takara, Dalian, China). Using the extracted total RNA as a template, single-stranded cDNA was synthesized through a reverse transcription kit (Takara). The prepared cDNA was used as a template with the primers listed in Appendix A to perform a Nested PCR reaction to amplify the ORF of *Lja-BCAP*. The amplification program was set as 94 °C for 5 min, 1 cycle; then 94 °C for 30 s, 49.6 °C for 30 s and 72 °C for 2.5 min, 30 cycles; finally, storage at 4 °C. The PCR products were detected by 1.5% agarose gel electrophoresis, and the target band was recovered from the gel and subcloned in the T vector (Takara), which was then verified by sequencing (Takara).

### 4.3. Homologous Sequence Comparison and Phylogenetic Analysis

The NCBI website was used to search the BCAP and BANK1 amino acid sequences of representative species in each class of vertebrates and invertebrates (Appendix A), and the BioEdit7.0 software was used for multiple sequence alignment. The conserved domains were predicted by the online tool SMART [40]. The NJ method was chosen to reconstruct the phylogenetic tree by using MEGA7.0 software. The Poisson correction method was used to calculate the evolutionary distance [41].

### 4.4. Conserved Motif Analysis

The conserved motifs of BCAP and BANK1 sequences mentioned above were analyzed by using an online tool Multiple Em for Motif Elicitation (https://meme-suite.org/meme/index.html, accessed on 20 April 2022). The parameters were set as follows: a single motif width is from 7 to 50; the total number of the motifs for searching is 20; the total number of sites in the primary sequence set where a single motif occurs is from 2 to 100 [42].

### 4.5. Recombinant Expression of Truncated Lja-BCAP

Primer 5.0 software was used to design primers with *EcoR*I and *Hind*Ⅲ restriction sites in the C-terminal of *Lja-BCAP* ORF that expresses relatively abundant epitopes (predicted by an online tool at http://tools.immuneepitope.org/main/bcell/, accessed on 11 October 2017). The target fragment was obtained by PCR reaction and subcloned into the pET-32a vector (Takara). The constructed recombinant expression vector was transformed into *E. coli* DH5α competent cells (Takara), and the recombinant plasmid was extracted and sequenced and verified by Takara. The verified truncated Lja-BCAP recombinant plasmid was transformed into the *E. coli* Rosetta (DE3) strain, and cultured to the logarithmic phase at 37 °C. Then the cultured strains were induced with IPTG (Shenggong, Shanghai, China) at a concentration of 1 mmol/L or 0.1 mmol/L at 30 °C and cultured 120 r/min for 4 h, and then the cells were collected. The bacterial cells were broken by ultrasound and then subjected to SDS-PAGE to detect the expression of recombinant proteins of Lja-BCAP (rLja-BCAP). The histidine tag (His-Tag) affinity gel chromatography method was used for purification.

### 4.6. Preparation and Titer Detection of Lja-BCAP Antibody

Rabbits (*Oryctolagus cuniculus*) are bred in our laboratory. Rabbit serum was taken as a negative control before immunization. During the first immunization, 600 μg of rLja-BCAP was phacoemulsified with an equal volume of Fischer’s complete adjuvant (Sigma, St. Louis, MO, USA), and then the rabbits were injected subcutaneously at multiple points on the back. After that, booster immunizations were carried out every two weeks with a half-dose of protein in Fischer’s incomplete adjuvant (Sigma, St. Louis, MO, USA). After the booster immunization, blood was collected from the ear vein every other week, and the antibody titer was detected by enzyme-linked immunosorbent assay (ELISA). Finally, blood was collected from the ear artery, incubated at 37 °C for 1 h, and centrifuged at 8000 r/min for 15 min to separate the serum. Antibodies are purified by affinity chromatography using Protein G gel columns (Beyotime, Shanghai, China). After the purified antibody was dialyzed, an equal amount glycerol was added, and the antibody was stored at −20 °C.

### 4.7. RNA Interference

The knockdown of Lja-BCAP mRNA was performed by RNAi technology [43]. Three pairs of siRNAs were designed and synthesized by Shanghai genepharma Co., Ltd. (Shanghai, China) as a siRNA pool for enhancing the interference effect (Table 1). The RNA in vivo transfection reagent Entranster^TM^-in-vivo was purchased from Engleen Biosystem Ltd. (Auckland, New Zealand). Lja-BCAP siRNA and Scrambled siRNA transfection complex were prepared according to the manufacturer’s instructions. Two milliliters of transfection complex contains 0.33 mg of siRNAs (0.11 mg for each), 0.25 mL of transfection reagent and 0.2 g of glucose. Briefly, mixed bacteria (*V. parahaemolyticus* and *S. agalactiae*) were used as antigens to immunize lampreys, and the first booster immunization was carried out seven days later. Lampreys treated in the same way with normal saline were set as blank control groups. Then after three days, each lamprey in control group was intraperitoneally injected with 200 μL transfection complex without siRNA, and those immunized with mixed-bacteria were injected with Lja-BCAP siRNA complex (set as RNAi group) and Scrambled siRNA complex (set as the negative control group), respectively. After 24 h of interference, lampreys were conducted second booster immunization with mixed bacteria (RNAi and negative control groups) or normal saline (blank control group). After 24 h stimulation, the lampreys were handled according to the method described in Section 4.1.

### 4.8. Quantitative Real-Time PCR (qPCR) Detection

The total RNA from peripheral blood lymphocytes, gills and supraneural myeloid bodies was extracted by RNAiso reagent, and cDNA was synthesized by reverse transcription with an AMV reverse transcription kit (TaKaRa). The Lja-BCAP primers designed in Appendix A were used for qPCR reaction with a SYBR^®^ Premix Ex Taq^TM^ premix kit (TaKaRa), and *gapdh* was used as an internal reference to analyze the differential expression of *Lja-BCAP* mRNA in the above-mentioned immune tissues after antigens stimulation [32]. Amplification conditions: pre-denaturation at 94 °C for 5 min; denaturation at 94 °C for 30 s, annealing at 60 °C for 30 s, 30 cycles, and storage at 4 °C. The relative expression levels of Lja-BCAP mRNA were calculated by using the 2^−ΔΔCT^ method [44].

### 4.9. Western Blotting Analysis

About 50 mg of lamprey tissue was homogenized in RIPA Lysis Buffer (Beyotime, Shanghai, China) plus protease inhibitors (Beyotime) in an ice bath and then centrifuged. The supernatant was added to the loading buffer and boiled for 8 min to prepare a protein sample. The protein samples were first separated by SDS-PAGE, then transferred to polyvinylidene fluoride (PVDF, Millipore) membrane for 75 min, and sealed with 5% skimmed milk powder for 3 h. Then the Lja-BCAP antibody (1:1000 *v*/*v*) was applied and incubated at 4 °C. After at least 4 h, the membrane was washed with 1 × TBST buffer (Tris buffer plus 0.05% Tween-20) for 10 min each time, five times in total. The secondary antibody (Goat anti-rabbit, Beyotime) was added at a ratio of 1:5000 and incubated at 37 °C for 1 h. the membrane was washed with 1 × TBST buffer, 10 min each time, four times in total. Finally, ECL luminescent solution (Beyotime) was added, and FluorChem Q multicolor fluorescence chemiluminescence imaging system (ProteinSimple, San Jose, CA, USA) was used to detect the fluorescence signal. The β-Actin Mouse mAb (ABclonal) was used to identify lamprey β-actin as internal control [32]. Densitometry data generated for WB to compare protein concentrations were analyzed with the software ImagePro 6.0.

### 4.10. Statistical Analysis

The obtained data were calculated by Graphpad prism 5.0 Software and the results were shown by mean ± standard deviation (m ± SD). The statistical difference between the two groups was analyzed by *t*-test, and *p* < 0.05 means significant, and *p* < 0.01 means extremely significant.

## 5. Conclusions

This work preliminarily characterized a BCAP homolog, Lja-BCAP, from lamprey. It is involved in anti-bacterial immunity by up-regulating expression and enhancing tyrosine phosphorylation levels in response to antigen challenges. It was also demonstrated that Lja-BCAP should mainly participate in the immune response of lamprey mediated by LPS. Of course, in-depth research is still needed in the future to reveal the relationships of Lja-BCAP with the signaling of lamprey VLRA^+^, VLRB^+^ and VLRC^+^ lymphocytes.

## Figures and Tables

**Figure 1 ijms-23-14449-f001:**
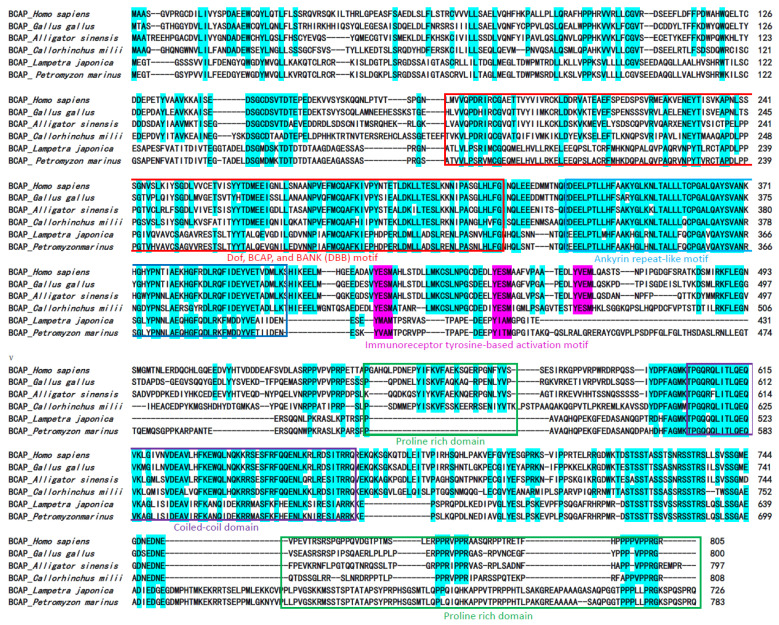
Sequence alignment of lamprey BCAP homologs with several typical BCAP molecules of higher vertebrates by using Clustal X program. The accession numbers of the sequences are listed in Appendix A. The identical amino acid residues among BCAP sequences are marked in light blue. The functional domains such as Dof, BCAP, and BANK (DBB) motif, anchor protein repeat motif, proline rich domain and coiled-coil domain are marked by red, blue, green and purple frames, respectively. The conserved immunoreceptor tyrosine-based activation motifs are indicated by pink letters.

**Figure 2 ijms-23-14449-f002:**
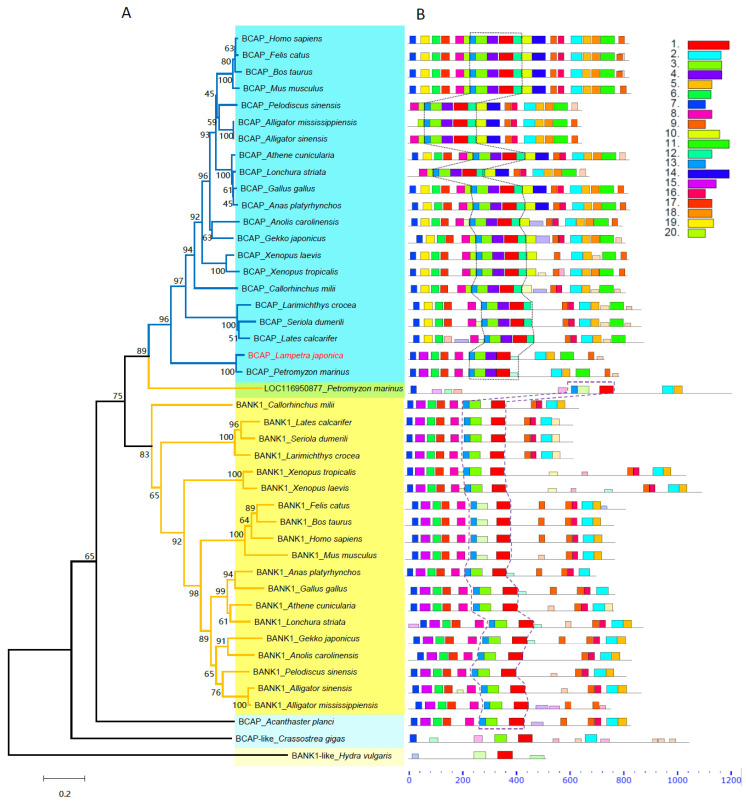
The phylogeny and conserved motif analysis of BCAP and BANK1 protein families. (**A**). The phylogenetical tree was reconstructed using the Neighbor-Joining method. The percentage of replicated trees in which the associated taxa clustered together in the bootstrap test (1000 replicates) is shown next to the branches. The tree is drawn to scale, with branch lengths in the same units as those of the evolutionary distances used to infer the phylogenetic tree. The BCAP and BANK1 clusters are shaded with light blue and yellow colors, respectively. The accession numbers of protein sequences are listed in Appendix A. (**B**). The conserved motif analysis was conducted by using an online tool Multiple Em for Motif Elicitation (https://meme-suite.org/meme/index.html, accessed on 20 April 2022). The 20 conserved motifs searched among 45 sequences are indicated with different colors and their sequences are shown in Appendix A. The unique conserved motif combination in BCAPs is marked by a dotted line, while in BANK1 it is marked by a dashed line.

**Figure 3 ijms-23-14449-f003:**
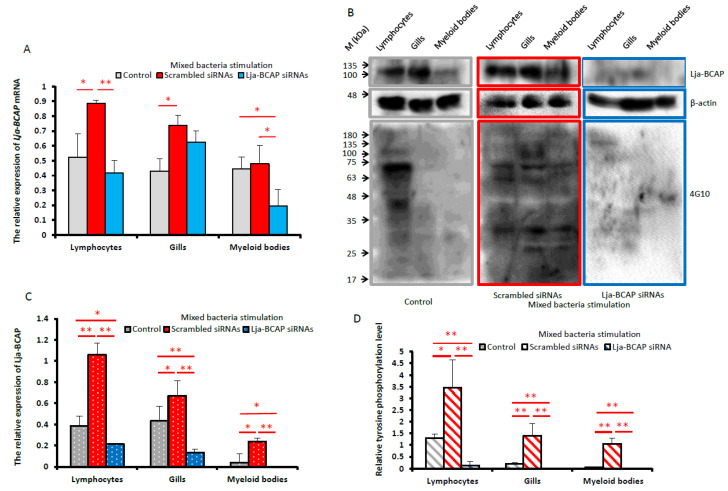
Investigation of the differential expression patterns of *Lja-BCAP* mRNA and protein after immune stimulation in immune-related tissues through RNA interference. (**A**). The differential expression patterns of *Lja-BCAP* mRNA after stimulation in immune-related tissues of lamprey treated with small interference RNA or not. Lampreys treated with Engleen transfection reagent, Lja-BCAP siRNAs transfection solution and scramble siRNAs transfection solution were set as control, RNAi and negative control groups, respectively. After 24 h interference, lampreys were stimulated by normal saline (control) or mixed bacteria, respectively. After 24 h stimulation, immune-related tissues such as peripheral blood lymphocytes (lymphocytes), gills and supraneural myeloid bodies (myeloid bodies) were isolated as described in Materials and Methods. (**B**). Western blotting method was used to detect the expression of Lja-BCAP and the tyrosine phosphorylation in immune-related tissues interfered with siRNA or not after mixed bacteria stimulation. (**C**,**D**). The statistical graphs of the relative expression of Lja-BCAP and the tyrosine phosphorylation levels calculated according to densitometry data generated from (**B**). The data shown in the figure are all carried out in three sets of parallel experiments for each group (n = 3). The experimental results are expressed as “mean ± SD”. All data are tested by *t*-test, * means significant difference (*p* < 0.05), ** means extremely significant difference (*p* < 0.01).

**Figure 4 ijms-23-14449-f004:**
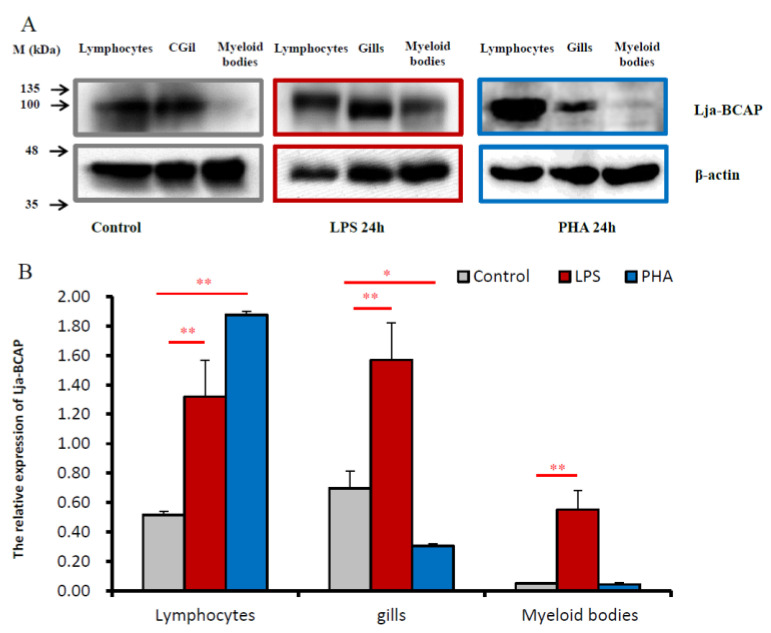
Relative expression levels of Lja-BCAP in immune-related tissues after LPS and PHA stimulation. (**A**) Western blotting method was used to detect the expression of Lja-BCAP in immune-related tissues after LPS and PHA stimulation, respectively. Lampreys were stimulated by normal saline, LPS or PHA for 24h, respectively, as described in Materials and Methods; (**B**) The statistical graph of the results of Western blotting. The data shown in the figure are all carried out in 3 sets of parallel experiments for each group (n = 3). The experimental results are expressed as “mean ± SD”. All data are tested by *t*-test, * means significant difference (*p* < 0.05), ** means extremely significant difference (*p* < 0.01).

**Table 1 ijms-23-14449-t001:** The siRNA sequences used in this study.

Symbol	Forward (5′-3′)	Reverse (5′-3′)
Lja-BCAP-546	CCGCAUCAUGUGUGGGCAATT	UUGCCCACACAUGAUGCGGTT
Scrambled-546	CCAAGCCUGGUGAUGGACUTT	AGUCCAUCACCAGGUUUGGTT
Lja-BCAP-891	GCUGCUCGCAGAUUCCUUATT	UAAGGAAUCUGCGAGCAGCTT
Scembled-891	CGAGACUCCGCAUAUUUCGTT	CGAAAUAUGCGGAGUCUCGTT
Lja-BCAP-1148	GCAAAUUCAUGGACGACUATT	UAGUCGUCCAUGAAUUUGCTT
Scembled-1148	CCAAGGAGGAUUUACCAUATT	UAUGGUAAAUCCUCCUUGGTT

## Data Availability

The data that support the findings of this study are available from the corresponding author upon reasonable request.

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
