# Peer review of "Molecular Characterization of a B Cell Adaptor for Phosphoinositide 3-Kinase Homolog in Lamprey (Lampetra japonica) and Its Function in the Immune Response"

_ijms, 2022, doi:10.3390/ijms232214449_

Round 1

Reviewer 1 Report

Review Manuscript: ijms-2035677

Molecular characterization of a BCAP homolog in lamprey (Lampetra japonica) and its function in immune response

This manuscript investigated an adaptor protein in a lamprey species at the nucleotide, the protein, and the functional level.  Much careful research was done, and the paper is worthy of publishing.  The subject may be narrow for some of the readership, but the methods and the importance to comparative immunology are of interest to many.  Please see a few constructive comments for improving the readability.

Write out BCAP in the title, and “…in the immune response”

Some of the acronyms in the manuscript are not defined.

A diagram(s) of the molecular cascade involved with cell signaling may be useful.

Line 77:  Do not start a sentence with a preposition.  Unclear sentence

Line 75:  “Based upon BCR, TCR, and the major histocompatibility complex, lamprey (jawless vertebrates) do not have an adaptive immune response.”

Line 101.  Citation missing for that statement.

Line 102.  This project is the preliminary analysis into the functional characterization of the role that Lja-BCAP.

Conclusions.  This work preliminarily characterized a BCAP homolog, Lja-BCAP, from lamprey.  It is involved in anti-bacterial immunity by up-regulating expression and ….”  Slight edits are needed here and throughout the manuscript.

Author Response

Dear reviewer:

Thank you for your constructive comments; we have revised our paper according to these valuable comments. The revisions are list below.

Comments and Suggestions for Authors

Review Manuscript: ijms-2035677

Molecular characterization of a BCAP homolog in lamprey (Lampetra japonica) and its function in immune response

This manuscript investigated an adaptor protein in a lamprey species at the nucleotide, the protein, and the functional level.  Much careful research was done, and the paper is worthy of publishing.  The subject may be narrow for some of the readership, but the methods and the importance to comparative immunology are of interest to many.  Please see a few constructive comments for improving the readability.

Comment 1

Write out BCAP in the title, and “…in the immune response”

Revision:

Molecular characterization of a B cell adaptor for phosphoinositide 3-kinase homolog in lamprey (Lampetra japonica) and its function in the immune response

Comment 2

Some of the acronyms in the manuscript are not defined.

Revisions:

Line 36:

Changed “immunoglobulins IgM and IgG3” to “immunoglobulin (Ig) M and IgG3”

Line 38:

Changed “After BCR binds to the ligand” to “After B cell receptor (BCR) binds to the ligand”

Line 46:

Changed “When the B cell receptor (BCR) signaling is triggered, ” to “When the BCR signaling is triggered,”

Line 53:

Changed “Src homology2 (SH2) domain” to “Src homology (SH) 2 domain”

Line 67:

Changed “transformation of CD8+ T cells into memory and effector T cells.” to “transformation of cluster of differentiation 8 (CD8) positive T cells into memory and effector T cells.”

Line 108:

Changed “It contains an ORF of 2181 bp,” to “It contains an open reading frame (ORF) of 2181 bp,”

Line 127:

Changed “we reconstructed the phylogenetic tree by NJ method” to “the phylogenetic tree was reconstructed by Neighbor-Joining (NJ) method“

Line 127:

Changed “with 45 BCAP and BANK1 sequences from 25 species” to “with 45 BCAP and B cell scaffold protein with ankyrin repeats 1 (BANK1) sequences from 25 species”

Line 152:

Changed “induced by IPTG, and then detected by SDS-PAGE (Figure S2A).” to “induced by Isopropyl β-D-thiogalactoside (IPTG), and then detected by sodium dodecylsulphate polyacrylamide gel electrophoresis (SDS-PAGE)”

Line 159:

Changed “according to the immunization method described above.” to “according to the immunization method described in Materials and Methods.”

Line 170:

Changed “The transcription and protein expression of Lja-BCAP were examined by qPCR” to “The transcription and protein expression of Lja-BCAP were examined by quantitative real-time PCR (qPCR)”

Line 208:

Changed “The WB results clearly showed that,” to “The Western blotting (WB) results clearly showed that,”

Line 347:

Changed “The knockdown of Lja-BCAP mRNA was performed by RNA interference (RNAi) technology [39].” to “The knockdown of Lja-BCAP mRNA was performed by RNAi technology [39]”

Line 348:

Changed “Three pairs of small interfering RNA (siRNA) were designed and synthesized” to “Three pairs of siRNAs were designed and synthesized”

Line 365:

Changed “for the next step of amplifying the open reading frame (ORF) of Lja-BCAP.” to “for the next step of amplifying the ORF of Lja-BCAP.”

Line 376:

Changed “to search the BCAP and B cell scaffold protein with ankyrin repeats 1 (BANK1) amino acid sequences” to “to search the BCAP and BANK1 amino acid sequences”

Line 380:

Changed “The Neighbor-Joining (NJ) method was chosen to reconstruct the phylogenetic tree” to “The NJ method was chosen to reconstruct the phylogenetic tree”

Line 398:

Changed “Then the cultured strains were induced with Isopropyl β-D-thiogalactoside (IPTG)” to “Then the cultured strains were induced with IPTG”

Line 401:

Changed “subjected to sodium dodecylsulphate polyacrylamide gel electrophoresis (SDS-PAGE) to detect the expression” to “subjected to SDS-PAGE to detect the expression”

Line 427:

Changed “4.9. Western blotting (WB) analysis” to “4.9. Western blotting analysis”

Comment 3

A diagram(s) of the molecular cascade involved with cell signaling may be useful.

Answer

The expert's opinion is very correct. A signal transduction pathway diagram will enable readers to understand BCR signal pathway more intuitively. However, if we draw by ourselves, it may involve copyright issues. Due to the budget, we did not draw the signal pathway diagram. Although it is not convenient, if readers are interested, they can obtain such a pathway diagram from the Internet.

Comment 4

Line 77:  Do not start a sentence with a preposition.  Unclear sentence

Revision:

Changed “But it has been found to form an alternative adaptive immune system in another way.” to “However, lampreys have been found to form an alternative adaptive immune system in another way.

Comment 5

Line 75:  “Based upon BCR, TCR, and the major histocompatibility complex, lamprey (jawless vertebrates) do not have an adaptive immune response.”

Revision:

Changed “Lamprey, as a representative of jawless vertebrates, has been proved without an adaptive immune system based on BCR, TCR and major histocompatibility complex (MHC).” to “Based upon BCR, TCR, and the major histocompatibility complex, lampreys (jawless vertebrates) do not have an adaptive immune response.

Comment 6

Line 101.  Citation missing for that statement.

Revision:

Changed “Recently, a molecule that is homologous to the higher vertebrate BCAP was isolated in Lampetra japonica (Lja-BCAP). ” to “In this study, we described the cloning, recombinant expression, antibody preparation and functional analysis of a BCAP homolog in Lampetra japonica (Lja-BCAP).”

Comment 7

Line 102.  This project is the preliminary analysis into the functional characterization of the role that Lja-BCAP.

Revision:

Changed “This article has conducted preliminary analyses in order to functional characterize the role of Lja-BCAP played in the immune response 103 of lamprey.” ” to “This project is the preliminary analysis into the functional characterization of the role that Lja-BCAP plays in the immune response.”

Comment 8

Conclusions.  This work preliminarily characterized a BCAP homolog, Lja-BCAP, from lamprey.  It is involved in anti-bacterial immunity by up-regulating expression and ….”  Slight edits are needed here and throughout the manuscript.

Revisions:

Line 449

Changed  “We have functional characterized a BCAP homolog, Lja-BCAP, from lamprey L. japonica. Lja-BCAP was proved to be involved in lamprey anti-bacterial immune response by up-regulating itself expression and enhancing tyrosine phosphorylation levels in response to antigen challenge.”

To  “This work preliminarily characterized a BCAP homolog, Lja-BCAP, from lamprey. It is involved in anti-bacterial immunity by up-regulating expression and enhancing tyrosine phosphorylation levels in response to antigen challenge.”

Line 126

Changed “In order to further understand the phylogenetic relationship of BCAP molecules, we reconstructed the phylogenetic tree by NJ method with 45 BCAP and BANK1 sequences from 25 species which are representatives of 6 vertebrate classes and 3 invertebrate classes (Figure 2A, Table S1).”

To “In order to further understand the phylogenetic relationship of BCAP molecules, the phylogenetic tree was reconstructed by Neighbor-Joining (NJ) method with 45 BCAP and B cell scaffold protein with ankyrin repeats 1 (BANK1) sequences from 25 species which are representatives of 6 vertebrate classes and 3 invertebrate classes (Figure 2A, Table S1).”

Line 262:

Changed “By searching through several genome databases, we find there are 2 BCAP homologous 262 sequences in sea lamprey (P. marinus).”

To “By searching through several genome databases, it was found that there are 2 BCAP homologous sequences in sea lamprey (P. marinus).”

Line 272:

Changed “Given that we could not find a sequence in lamprey genome with high homology to vertebrates’ BANK1 in these genomic databases available,”

To “Given that no sequence could be found in lamprey genome with high homology to vertebrates’ BANK1 in these genomic databases available”

Line 314:

Changed  “we have tested the expression profiles of Lja-BCAP in these immune related tissues under LPS and PHA stimulation.”

To  “the expression profiles of Lja-BCAP were tested in these immune related tissues under LPS and PHA stimulation.”

Line 452:

Changed  “We also demonstrated that Lja-BCAP should mainly partici-452 pate in the immune response of lamprey mediated by LPS.”

To  “It was also demonstrated that Lja-BCAP should mainly participate in the immune response of lamprey mediated by LPS.”

Reviewer 2 Report

In this study, Chai et al. identified and cloned cDNA for B cell adaptor for phosphoinositide 3-kinase (BCAP) from Lampetra japonica. Lja BCAP's 200 amino acids at its C-terminus were expressed in E coli and used to immunize rabbits and generate polyclonal antibodies. The effects of RNAi on Lja-BCAP were studied by Western Blot and quantitative PCR. This manuscript follows standard methods for this type of research. Methods and results should be restructured and presented in a more logical sequence in the paper in order to make it easier for the reader to comprehend.

Major Comments

Major Comment 1:  Large protein Size difference on BCAP protein among closely related Lamprey genera? BCAP protein seems to be fairly conserved in size, around 800 amino acids (Human BCAP is 805 aa, chicken 800 and Reptile 797 aa) . But why does BCAP protein the two lamprey genera Lampetra and Petromyzon which belong to the same family Petromyzontidae have huge variation in the amino acid length (around 60 amino acid difference). Did the authors consider the case of alternative splicing and whether the 726 aa form identified be a shorter isoform of the canonical BCAP variant. It could be possible that there could be other splice forms coding for similar size around Petromyzon BCAP (783 aa).

The authors have used only a single amplicon from supraneural myeloid bodies for cloning the BCAP cDNA fragment. Considering the possibility of alternative splicing, and high level of BCAP mRNA expression noted in gills, lymphocytes it would be interesting to amplify and clone BCAP amplicons from these tissues as well and sequence the clones.

Major comment 2 : The authors should consider a major logical reorganisation of the sequence of presentation of the results and materials and methods. The materials & methods start with the immunization and RNAi based gene silencing even before the identification and cloning of the Lja BCAP. The sequence of the methods and results are quite different and is very confusing for the reader to comprehend on the method followed for a particular result.

For the results presented in Section 2.5, the corresponding method is not seen in Materials & methods

Major Comment 3: Considering the authors had produced antibodies against Lja BCAP, why didn't the authors use immunohistochemistry to demonstrate the expression of Lja BCAP in control and RNAi gene silenced lamprey tissues?

Major comment 4 : Details lacking in Section 4.2. RNA interference :. For in-vivo RNAi for LjaBCAP, the authors are requested to provide more details regarding the amount of siRNA used, the dose/weight, the ratio of RNA to transfection reagent, etc, with appropriate references.

Major comment 5: Figure 3B- Please provide a bar chart figure with appropriate statistical significance annotations for the levels of tyrosine phosphorylation following the knockdown of Lja-BCAP siRNA/scrambled siRNA/control in Fig 3B.Due to the low quality of the blot for 4G10 shown in Figure 3B, it would be challenging to assess tyrosine phosphorylation level from the blot image alone.

Minor comments :

B cell /B-cell : authors are requested to stick to a single standard format throughout the manuscript.

Line 68 Use italics for all binomials -  Listeria monocytogenes

Line 107  Nest-PCR  should be Nested PCR

Line 152 – Correct to Rosetta, there are many types of Rosetta E. coli expression strains. Kindly specify the full name of Rosetta strains used for expression

Line 155- Water soluble protein ?  Kindly check the usage and correct. Solubility/Insolubility of recombinant proteins refers to the expression of recombinant protein inside E coli cytoplasm/ or as insoluble aggregates or inclusion bodies.

Author Response

Dear reviewer:

Thank you for your constructive comments; we have revised our paper according to these valuable comments. The revisions are list below.

Comments and Suggestions for Authors

In this study, Chai et al. identified and cloned cDNA for B cell adaptor for phosphoinositide 3-kinase (BCAP) from Lampetra japonica. Lja BCAP's 200 amino acids at its C-terminus were expressed in E coli and used to immunize rabbits and generate polyclonal antibodies. The effects of RNAi on Lja-BCAP were studied by Western Blot and quantitative PCR. This manuscript follows standard methods for this type of research. Methods and results should be restructured and presented in a more logical sequence in the paper in order to make it easier for the reader to comprehend.

Major Comments

Major Comment 1:  Large protein Size difference on BCAP protein among closely related Lamprey genera? BCAP protein seems to be fairly conserved in size, around 800 amino acids (Human BCAP is 805 aa, chicken 800 and Reptile 797 aa) . But why does BCAP protein the two lamprey genera Lampetra and Petromyzon which belong to the same family Petromyzontidae have huge variation in the amino acid length (around 60 amino acid difference). Did the authors consider the case of alternative splicing and whether the 726 aa form identified be a shorter isoform of the canonical BCAP variant. It could be possible that there could be other splice forms coding for similar size around Petromyzon BCAP (783 aa).

The authors have used only a single amplicon from supraneural myeloid bodies for cloning the BCAP cDNA fragment. Considering the possibility of alternative splicing, and high level of BCAP mRNA expression noted in gills, lymphocytes it would be interesting to amplify and clone BCAP amplicons from these tissues as well and sequence the clones.

Answer for comment 1:

The reviewer’s comment is reasonable. Surely, there are 4 Petromyzon BCAP sequences in NCBI data base. The one (AY152674, 783aa) cited by us was first characterized by Uinuk-Ool et al. at 2002, the other three isoforms (XP_032805722, 798aa; XP_032805723, 786aa; XP_032805724, 738aa) were derived from a genomic sequence (NC_046076.1) annotated using gene prediction method at 2020. Lja-BCAP (726aa) is 98.34% identical to isoform XP_032805724 (Figure A below). Considering these 3 isoforms are predicted form genome sequence but not verified by cloning sequence, we choose sequence AY152674 for sequence alignment and evolutional tree construction.

               ....|....| ....|....| ....|....| ....|....| ....|....| ....|....| ....|....| 70

Lja-BCAP       MEGTGSSSVV ILFDENGYQW GDYMVQLLKA KQTCLRCRKI SLDGTPLSRG DSSAIGTASC RLLILTDGLM

XP_032805724   MEGTGSYPVV ILFDENGYEW GDYMVQLLKA KQTCLRCRKI SLDGKPLSRG DSSAIGTASC RVLILTAGLM

AY152674       MEGTGSYPVV ILFEEDGYEW GDYMVQLLKV RQTCLRCRKI SLDGKPLSRG DSSAIGTASC RVLILTAGLM

XP_032805723   MEGTGSYPVV ILFDENGYEW GDYMVQLLKA KQTCLRCRKI SLDGKPLSRG DSSAIGTASC RVLILTAGLM

XP_032805722   MEGTGSYPVV ILFDENGYEW GDYMVQLLKA KQTCLRCRKI SLDGKPLSRG DSSAIGTASC RVLILTAGLM

Clustal Cons   ****** .** ***:*:**:* *********. :********* ****.***** ********** *:**** ***

               ....|....| ....|....| ....|....| ....|....| ....|....| ....|....| ....|....| 140

Lja-BCAP       EGLTDWPMTR DLLKLLVPPK SVLLLLCGVS EEDAQGLLAA LHVSHRWTIL SCESAPESFV ATITDIVTEG

XP_032805724   EGLTDWPMSR DLLKSLVPPK SVLLLLCGVS EEDAQGLLVA LHVSHRWKIL SCESAPENFV ATITDIVTEG

AY152674       EGLTDWPMSR DLLKSLVPPK SVLLLLCGVS EEDAQGLLVA LHVSHRWKIL SCGSAPENFV ATITDIVTEG

XP_032805723   EGLTDWPMSR DLLKSLVPPK SVLLLLCGVS EEDAQGLLVA LHVSHRWKIL SCESAPENFV ATITDIVTEG

XP_032805722   EGLTDWPMSR DLLKSLVPPK SVLLLLCGVS EEDAQGLLVA LHVSHRWKIL SCESAPENFV ATITDIVTEG

Clustal Cons   ********:* **** ***** ********** ********.* *******.** ** ****.** **********

               ....|....| ....|....| ....|....| ....|....| ....|....| ....|....| ....|....| 210

Lja-BCAP       GTADELDSGM DSKTDTDTDT AAGDAGESSA SPRGNATLVL PSRIMCGQQM ELHVLLRKEL EEQPSLTCRF

XP_032805724   GTADELDSGM DMKTDTDTDT AAGEAGASSA SPRGSATVVL PSRVMCGEQM ELHVLLRKEL EEQPSLACRF

AY152674       -TADELDSGM DMKTDTDTDT AAGEAGASSA SPRGSATVVL PSRVMCGEQM ELHVLLRKEL EEQPSLACRF

XP_032805723   GTADELDSGM DMKTDTDTDT AAGEAGASSA SPRGSATVVL PSRVMCGEQM ELHVLLRKEL EEQPSLACRF

XP_032805722   GTADELDSGM DMKTDTDTDT AAGEAGASSA SPRGSATVVL PSRVMCGEQM ELHVLLRKEL EEQPSLACRF

Clustal Cons    ********* * ******** ***:** *** ****.**:** ***:***:** ********** ******:***

               ....|....| ....|....| ....|....| ....|....| ....|....| ....|....| ....|....| 280

Lja-BCAP       MHKNQPALQV PAQRVNPYTL RCTAPDLPPG IVQVAVCSAG AVRESTSLTY YTALQEVGDI LGDVNNPIAF

XP_032805724   MHKDQPALQV PAQRVNPYTV RCTAPDLPPG TVHVAVCSAG VVRESTSLTY YTALQEVGNI LEDVNNPIAF

AY152674       MHKDQPALQV PAQRVNPYTV RCTAPDLPPG TVHVAVCSAG VVRESTSLTY YTALQEVGNI LEDVNNPIAF

XP_032805723   MHKDQPALQV PAQRVNPYTV RCTAPDLPPG TVHVAVCSAG VVRESTSLTY YTALQEVGNI LEDVNNPIAF

XP_032805722   MHKDQPALQV PAQRVNPYTV RCTAPDLPPG TVHVAVCSAG VVRESTSLTY YTALQEVGNI LEDVNNPIAF

Clustal Cons   ***:****** *********: **********  *:******* .********* ********:* * ********

               ....|....| ....|....| ....|....| ....|....| ....|....| ....|....| ....|....| 350

Lja-BCAP       MCQAFKIEPH DPERLDMLLA DSLRENLPAS NVHLFGNHQL SNNNTQREEE LPTLMHFAAK HGLRNLTALL

XP_032805724   MCQAFKIEPH DPERLDMLLA DSLRENLPAS NLHLFGNHQL SNTNTQREEE LPTLLHFAAK HGLHNLTALL

AY152674       MCQAFKIEPH DPERLDMLLA DSLRENLPAS NLHLFGNHQL SNTNTQREEE LPTLLHFAAK HGLHNLTALL

XP_032805723   MCQAFKIEPH DPERLDMLLA DSLRENLPAS NLHLFGNHQL SNTNTQREEE LPTLLHFAAK HGLHNLTALL

XP_032805722   MCQAFKIEPH DPERLDMLLA DSLRENLPAS NLHLFGNHQL SNTNTQREEE LPTLLHFAAK HGLHNLTALL

Clustal Cons   ********** ********** ********** *:******** **.******* ****:***** ***:******

               ....|....| ....|....| ....|....| ....|....| ....|....| ....|....| ....|....| 420

Lja-BCAP       FQCPGAVQAY SVANRSGLYP NNLAEQHGFQ DLRKFMDDYV EAIIDENESE YMAMTPSRVA STPAPEDEEP

XP_032805724   FQCPGAVQAY SVANRSGLYP NNLAEQHGFQ DLRKFMDDYV ETIIDENESK YVAMTPCRVP PTPAPEDEEP

AY152674       FQCPGAVQAY SVANRSGLYP NNLAEQHGFQ DLRKFMDDYV ETIIDENESK YVAMTPCRVP PTPAPEDEEP

XP_032805723   FQCPGAVQAY SVANRSGLYP NNLAEQHGFQ DLRKFMDDYV ETIIDENESK YVAMTPCRVP PTPAPEDEEP

XP_032805722   FQCPGAVQAY SVANRSGLYP NNLAEQHGFQ DLRKFMDDYV ETIIDENESK YVAMTPCRVP PTPAPEDEEP

Clustal Cons   ********** ********** ********** ********** *:*******: *:****.**. .*********

               ....|....| ....|....| ....|....| ....|....| ....|....| ....|....| ....|....| 490

Lja-BCAP       YIAMGPGITE ---------- ---------- ---------- ---------- ---------- ----------

XP_032805724   YITMGPGITE ---------- ---------- ---------- ---------- ---------- ----------

AY152674       YITMGPGITA KQQSLRALGR ERAYCGVPLP SDPFGLFGLT HSDASLRNLL EGTTQEMQSG PPKARPANTE

XP_032805723   YITMGPGITA KQQSLRALGR ERAYCGVPLP SDPFGLFGLT HSDASLRNLL EGTTQEMQSG PPKARPANTE

XP_032805722   YITMGPGITA KQQSLRALGR ERAYCGVPLP SDPFGLFGLT HSDASLRNLL EGTTQEMQSG PPKARPANTE

Clustal Cons   **:******                                                                    

               ....|....| ....|....| ....|....| ....|....| ....|....| ....|....| ....|....| 560

Lja-BCAP       ERSQQNLPKR ASLKPTRSPP AVAQHQPEKG FEDASANQGP TRDHFAGMKT PGQQQLITLQ EQVKAGLISI

XP_032805724   ERSQQNWPKR ASLKPARSPP AVAQHQPEKG FEDASANQDP THDHFAGMKT PGQQQLITLQ EQVKAGLISI

AY152674       ERSQQNWPKR ASLKPARSPP AVAQHQPEKG FEDASANQDP AHDHFAGMKT PGQQQLITLQ EQVKAGLISI

XP_032805723   ERSQQNWPKR ASLKPARSPP AVAQHQPEKG FEDASANQDP THDHFAGMKT PGQQQLITLQ EQVKAGLISI

XP_032805722   ERSQQNWPKR ASLKPARSPP AVAQHQPEKG FEDASANQDP THDHFAGMKT PGQQQLITLQ EQVKAGLISI

Clustal Cons   ****** *** *****:**** ********** ********.* ::******** ********** **********

               ....|....| ....|....| ....|....| ....|....| ....|....| ....|....| ....|....| 630

Lja-BCAP       DEAVIRFKAN QIDEKRRMAS FKFHEENLKS IRESIARRKK EPSPRQPDLK EDIPVGLYES LPSKEVPFPS

XP_032805724   DEAVIRFKAN QIDEKRRMAS FKFHEENLKN IRESIARRKK EPSLKQPDLN EDIAVGLYES LPSKEVPLPS

AY152674       DEAVIRFKAN QIDEKRRMAS FKFHEENLKN IRESIARRKK EPSLKQPDLN EDIAVGLYES LPSKEVPLPS

XP_032805723   DEAVIRFKAN QIDEKRRMAS FKFHEENLKN IRESIARRKK EPSLKQPDLN EDIAVGLYES LPSKEVPLPS

XP_032805722   DEAVIRFKAN QIDEKRRMAS FKFHEENLKN IRESIARRKK EPSLKQPDLN EDIAVGLYES LPSKEVPLPS

Clustal Cons   ********** ********** *********. ********** *** :****: ***.****** *******:**

               ....|....| ....|....| ....|....| ....|....| ....|....| ....|....| ....|....| 700

Lja-BCAP       QGAFRHRPMR DSTSSTTSSV SSRSSTRSLQ SLSSGAEADI E--------- ---DGEGDMP HTMKEKRRTS

XP_032805724   QGAFRHRPMR DSTSSTTSSV SSRSSTRSLQ SLSSGAEADI EVTDILTSQA SEGDGEGDMP HTMKEKRRTS

AY152674       QGAFRHRPMR DSTSSTTSSV SSRSSTRSLQ SLSSGAEADI E--------- ---DGEGDMP HTMKEKRRTS

XP_032805723   QGAFRHRPMR DSTSSTTSSV SSRSSTRSLQ SLSSGAEADI E--------- ---DGEGDMP HTMKEKRRTS

XP_032805722   QGAFRHRPMR DSTSSTTSSV SSRSSTRSLQ SLSSGAEADI EVTDILTSQA SEGDGEGDMP HTMKEKRRTS

Clustal Cons   ********** ********** ********** ********** *             ******* **********

               ....|....| ....|....| ....|....| ....|....| ....|....| ....|....| ....|....| 770

Lja-BCAP       ELPMLEKKCV PPLPVGSKKM SSTSPTATAP SYPRPHSGSM TLQPPQIQHK APPVTPRPPH TLSAKGREAP

XP_032805724   EPPMLGKKYV PPLPVGSKRM SSTSPTATAP SYPRPHSGSM TLQPLQIQHK APPVTPRPPH TLPAKGQEAP

AY152674       EPPMLGKNYV PLLPVGSKRM SSTSPTATAP SYPRPHSGSM TLQPLQIQHK APPVTPRPPH TLPAKGRE--

XP_032805723   EPPMLGKKYV PPLPVGSKRM SSTSPTATAP SYPRPHSGSM TLQPLQIQHK APPVTPRPPH TLPAKGQEAP

XP_032805722   EPPMLGKKYV PPLPVGSKRM SSTSPTATAP SYPRPHSGSM TLQPLQIQHK APPVTPRPPH TLPAKGQEAP

Clustal Cons   * *** *: * * ******:* ********** ********** **** ***** ********** **.***:*  

               ....|....| ....|....| ....|... 790

Lja-BCAP       AAAGASAQPG GTPPPLLPRG KSPQSPRQ

XP_032805724   AAAAASAQPR GTPPPLLPRG KSPQSPRQ

AY152674       AAAAASAQPG GTPPPLLPRG KSPQSPRQ

XP_032805723   AAAAASAQPR GTPPPLLPRG KSPQSPRQ

XP_032805722   AAAAASAQPR GTPPPLLPRG KSPQSPRQ

Clustal Cons   ***.*****  ********** ********

                Figure A The sequence alignment of Lja-BCAP with sea lamprey BCAP variants

     The diversified parts are marked by red color. The predicted epitope sequences are marked with green letters.

At first, we don’t know where we can capture cDNA sequence of Lja-BCAP, we have prepared cDNA templates from lymphocytes, gills and supraneural myeloid bodies stimulated by mixed-bacteria or normal saline for PCR. Fortunately, a single band appeared after nested-PCR application only in the sample made from supraneural myeloid bodies stimulated by mixed-bacteria. Because there may be various reasons why BCAP is not amplified in lymphocytes and gills, we have not continued to try to amplify BCAP in lymphocytes and gills based on the thinking of having more than no. Fortunately, the 200aa-fragment of Lja-BCAP in c-terminal region is 91.5% identical to sea lamprey BCAP AY152674 (Figure A), and the differences among these variants are not remarkable in this position. Thus the polyclonal antibodies we generated should be effective in detecting these variants.

Thanks a lot for reviewer provide us this important information, this suggestion give us a clue for future studying the alternative splicing isoforms of Lja-BCAP and their expression profiles and functional roles in immune related tissues. 

Major comment 2 : The authors should consider a major logical reorganisation of the sequence of presentation of the results and materials and methods. The materials & methods start with the immunization and RNAi based gene silencing even before the identification and cloning of the Lja BCAP. The sequence of the methods and results are quite different and is very confusing for the reader to comprehend on the method followed for a particular result.

For the results presented in Section 2.5, the corresponding method is not seen in Materials & methods

Revisions for comment 2:

We have changed the sequence of methods according to reviewer’s comment.

Line 361:

4.3 changed to 4.2

Line 375:

4.4 changed to 4.3

Line 383:

4.5 changed to 4.4

Line 389:

4.6 changed to 4.5

Line 404:

4.7 changed to 4.6

Line 346:

4.2 changed to 4.7

Line 340

We have added the method for preparing LPS and PHA solution in 4.1.

Original:

“Lampreys were immunized according to the previous description [30] after 1 week of acclimatization in a constant temperature water tank. Two aquatic bacteria Streptococcus agalactiae and Vibrio parahemolyticus were chosen as the representatives of Gram-positive (G+) and Gram-negative (G-) bacteria for preparing inactivated antigens, respectively. They were inactivated by acetaldehyde first and then mixed together in normal saline at a concentration of 1×107 CFU/ml for each strain.”

Revised:

“Two aquatic bacteria Streptococcus agalactiae and Vibrio parahemolyticus were chosen as the representatives of Gram-positive (G+) and Gram-negative (G-) bacteria for preparing inactivated antigens, respectively. They were inactivated by acetaldehyde first and then mixed together in normal saline at a concentration of 1×107 CFU/ml for each strain. LPS or PHA powder were dissolved in phosphate buffered saline (PBS) buffer (0.01M, pH7.4), respectively, at a concentration of 1mg/ml.  After 1 week of acclimatization in a constant temperature water tank, lampreys in each group were immunized with corresponding antigen solution (100μl for each) through intraperitoneal injection according to the previous description [30].”

Major Comment 3: Considering the authors had produced antibodies against Lja BCAP, why didn't the authors use immunohistochemistry to demonstrate the expression of Lja BCAP in control and RNAi gene silenced lamprey tissues?

Answer to comment 3:

The immunohistochemistry is an intuitive method to observe the protein expression. Unfortunately, our frozen slicer was damaged last year. Due to the strict anti-epidemic policy, the parts could not be imported smoothly, and the price was much higher than normal, so the repair was delayed until now. In addition, our students cannot enter or leave the campus without official permit, nor can they enter or leave the campus of other universities at will, which makes this experiment impossible. Considering that we only published a short communication paper this time, we have to give up this part content. In the future, after we have successfully prepared high affinity VLRB, VLRA and VLRC antibodies, we are going to knock down these molecules and detect the expression of BCAP in immune related tissues. At that time, immunohistochemistry results are absolutely necessary part to be included.

Major comment 4 : Details lacking in Section 4.2. RNA interference :. For in-vivo RNAi for LjaBCAP, the authors are requested to provide more details regarding the amount of siRNA used, the dose/weight, the ratio of RNA to transfection reagent, etc, with appropriate references.

Revisions for comment 4:

Original:

“The knockdown of Lja-BCAP mRNA was performed by RNAi technology [39]. Three pairs of siRNAs were designed and synthesized by Shanghai genepharma Co., Ltd. (Shanghai, China) as a siRNA pool for enhancing the interference effect (Table 1). The RNA in vivo transfection reagent EntransterTM-in-vivo was purchased from Engleen Biosystem Ltd. (Auckland, New Zealand). Briefly, mixed bacteria (V. parahaemolyticus and S. agalactiae) were used as antigens to immunize lampreys, and the first booster immunization was carried out seven days later. Then after three days, Lja-BCAP siRNA and Scrambled siRNA transfection complex were prepared according to the manufacturer's instructions, and lampreys were intraperitoneally injected with transfection reagent (control group), Lja-BCAP siRNA complex (RNAi group), and Scrambled siRNA complex (negative control group), respectively. After 24 hours of interference, the lampreys were conducted second booster immunization with mixed bacteria. After 24 hours, the lampreys were treated according to the method described above.”

Revised:

“The knockdown of Lja-BCAP mRNA was performed by RNAi technology [39]. Three pairs of siRNAs were designed and synthesized by Shanghai genepharma Co., Ltd. (Shanghai, China) as a siRNA pool for enhancing the interference effect (Table 1). The RNA in vivo transfection reagent EntransterTM-in-vivo was purchased from Engleen Biosystem Ltd. (Auckland, New Zealand). Lja-BCAP siRNA and Scrambled siRNA transfection complex were prepared according to the manufacturer's instructions. Two milliliter of transfection complex contains 0.33mg of siRNAs (0.11mg for each), 0.25ml of transfection reagent and 0.2g of glucose. Briefly, mixed bacteria (V. parahaemolyticus and S. agalactiae) were used as antigens to immunize lampreys, and the first booster immunization was carried out seven days later. Lampreys treated in the same way with normal saline were set as blank control group. Then after three days, each lamprey in control group was intraperitoneally injected with 200 μl transfection complex without siRNA, and those immunized with mixed-bacteria were injected with Lja-BCAP siRNA complex (set as RNAi group) and Scrambled siRNA complex (set as negative control group), respectively. After 24 hours of interference, lampreys were conducted second booster immunization with mixed bacteria (RNAi and negative control groups) or normal saline (blank control group). After 24 hours stimulation, the lampreys were handled according to the method described in section 4.1.”

Major comment 5: Figure 3B- Please provide a bar chart figure with appropriate statistical significance annotations for the levels of tyrosine phosphorylation following the knockdown of Lja-BCAP siRNA/scrambled siRNA/control in Fig 3B.Due to the low quality of the blot for 4G10 shown in Figure 3B, it would be challenging to assess tyrosine phosphorylation level from the blot image alone.

Revisions for comment 5:

We have added a bar chart figure in Fig 3 as below:

Figure 3. Investigation of the differential expression patterns of Lja-BCAP mRNA and protein after immune stimulation in immune related tissues through RNA interference. (A). The differential expression patterns of Lja-BCAP mRNA after stimulation in immune-related tissues of lamprey treated with small interference RNA or not. Lampreys treated with Engleen transfection reagent, Lja-BCAP siRNAs transfection solution and scramble siRNAs transfection solution were set as control, RNAi and negative control groups, respectively. After 24h interference, lampreys were stimulated by normal saline (control) or mixed bacteria, respectively. After 24h stimulation, immune related tissues such as peripheral blood lymphocytes (lymphocytes), gills and supraneural myeloid bodies (myeloid bodies) were isolated as described in Materials and Methods. (B). Western blotting method was used to detect the expression of Lja-BCAP and the tyrosine phosphorylation in immune-related tissues interfered with siRNA or not after mixed bacteria stimulation. (C,D). The statistical graphs of the relative expression of Lja-BCAP and the tyrosine phosphorylation levels calculated according to densitometry data generated from (B). The data shown in the figure are all carried out in 3 sets of parallel experiments for each group (n=3). The experimental results are expressed as "mean ± SD". All data are tested by t-test, * means significant difference (P<0.05), ** means extremely significant difference (P<0.01).

Minor comment 1: B cell /B-cell : authors are requested to stick to a single standard format throughout the manuscript.

Revision:

Line 93

Changed “B-cell” to “B cell”

Minor comment 2:  Line 68 Use italics for all binomials -  Listeria monocytogenes

Revision:

Line68

Changed “Listeria monocytogenes” to “Listeria monocytogenes

Minor comment 3: Line 107  Nest-PCR  should be Nested PCR

Revision:

Line 107

Changed “Nest-PCR” to “Nested PCR”

Line 370

Changed “nested PCR” to “Nested PCR”

Minor comment 4: Line 152 – Correct to Rosetta, there are many types of Rosetta E. coli expression strains. Kindly specify the full name of Rosetta strains used for expression

Revision:

Line 152

Changed “E. coli Rosstta competent cells” to “E. coli Rosstta (DE3) competent cells”

Line 397

Changed “E. coli Rosetta strain” to “E. coli Rosstta (DE3) strain”

Minor comment 5: Line 155- Water soluble protein ?  Kindly check the usage and correct. Solubility/Insolubility of recombinant proteins refers to the expression of recombinant protein inside E coli cytoplasm/ or as insoluble aggregates or inclusion bodies.

Revision:

Line 155

Changed “rLja-BCAP was a water soluble protein” to “the solubility of rLja-BCAP was high”

Round 2

Reviewer 2 Report

In reviewing the updated manuscript and the author's response to the review comments, I feel the paper has improved by incorporating the initial suggestions and explaining the queries. Methods were presented more logically and the requested details were provided. In figure 3, a bar chart was also provided for tyrosine phosphorylation levels as requested. Other minor errors were also corrected. Recommend the revised manuscript for publication.

NOTE: Pls correct Line - 164 - Pls correct Rosstta(DE3)  to Rosetta (DE3)